# Readiness of primary care centres for a community-based intervention to prevent and control noncommunicable diseases in the Caribbean: A participatory, mixed-methods study

Reeta Gobin[1], Troy Thomas[2], Sharlene Goberdhan[1]*, Manoj Sharma[1], Robert Nasiiro[3], Rosana Emmanuel[4], Madan Rambaran[5], Shelly McFarlane[6], Christelle Elia[7], Davon Van-Veen[1], Ishtar Govia[6], Tiffany Palmer[6], Ursula Read[8], J. Kennedy Cruickshank[4], T. Alafia Samuels[6], Rainford Wilks[6], Seeromanie Harding[4]

1 College of Medical Sciences, University of Guyana, Georgetown, Guyana, 2 Faculty of Natural Sciences, University of Guyana, Georgetown, Guyana, 3 School of Medicine, Ross University, Roseau, Dominica, 4 School of Life Course and Population Sciences, King's College London, London, United Kingdom, 5 Institute of Health Science Education, Georgetown Public Hospital Corporation, Georgetown, Guyana, 6 Caribbean Institute for Health Research, University of the West Indies, Kingston, Jamaica, 7 IQVIA, London, United Kingdom, 8 University of Essex, Essex, United Kingdom

* sharlene.goberdhan@uog.edu.gy

## Abstract

### Introduction

Epidemiological transition to NCDs is a challenge for fragile health systems in the Caribbean. The Congregations Taking Action against NCDs (CONTACT) Study intervention proposes that trained health advocates (HAs) from places of worship (PoWs), supervised by nurses at nearby primary healthcare centres (PHCs), could facilitate access to primary care among vulnerable communities. Drawing on participatory and systems thinking, we explored the capacity of local PHCs in three Caribbean countries to support this intervention.

### Methods

Communities in Jamaica (rural, urban), Guyana (rural) and Dominica (Indigenous Kalinago Territory) were selected for CONTACT because of their differing socio-economic, cultural, religious and health system contexts. Through mixed-method concept mapping, we co-developed a list of perceived actionable priorities (possible intervention points ranked highly for feasibility and importance) with 48 policy actors, healthcare practitioners and civic society representatives. Guided in part by the concept mapping findings, we assessed the readiness of 12 purposefully selected PHCs for the intervention, using a staff questionnaire and an observation checklist to identify enablers and constrainers.

### Results

Concept mapping illustrated stakeholder optimism for the intervention, but revealed perceptions of inadequate primary healthcare service capacity, resources and staff training to

**Data Availability Statement:** All relevant data are within the manuscript and its Supporting Information files.

**Funding:** This work is being funded by the Medical Research Council, grant references MR/N015959/1, MR/S003444/1, MR/Y009983/1, MR/X009777/1 and MR/X003078/1, awarded to SH. https://www.ukri.org/councils/mrc/. The funders had no role in study design, data collection and analysis, decision to publish, or preparation of the manuscript.

**Competing interests:** The authors have declared that no competing interests exist.

support implementation. Readiness assessments of PHCs identified potential enablers and constrainers that were consistent with concept mapping results. Staff support was evident. Constraints included under-staffing, which could hinder supervision of HAs; and inadequate essential NCD medicines, training in NCDs and financial and policy support for embedding community interventions. Despite a history of socio-political disadvantage, the most enabling context was found in the Kalinago Territory, where ongoing community engagement activities could support joint development of programmes between churches and PHCs.

## Conclusion

Multi-sectoral stakeholder consultation and direct PHC assessments revealed viability of the proposed POW-PHC partnership for NCD prevention and control. However, structural and policy support will be key for implementing change.

## Introduction

The World Health Organization's (WHO) 2013–20 Global Non-communicable Disease (NCD) Action Plan calls for a 25% reduction in NCD-related premature mortality by 2025 [1]. The United Nations Sustainable Development Goal (SDG) targets a one-third reduction by 2030 [2]. Despite regional and national frameworks and agendas, there is limited evidence that the Caribbean will meet these targets. The region has the highest NCD related mortality in all of the Americas, accounting for 62%-80% of all premature deaths (30–70 years) across the Caribbean Commonwealth countries [3]. The 2016 Port of Spain Declaration Evaluation found that heart attacks, stroke and diabetes cause most premature deaths, followed by cancers [4]. Hypertension is the leading risk factor for death, and the prevalence of type 2 diabetes is twice the global level, with rates as high as 15% in some territories [5]. Such epidemiological transition to NCDs, in context of viral infection outbreaks (previously HIV, recurrent dengue, zika, chikungunya and recently COVID-19) is an enormous challenge for the region's health systems. These are under-resourced and need structural and policy reform to move from reactive to pro-active, preventive actions.

It is critical that approaches to NCD prevention and control include translation of knowledge about social determinants into effective, large-scale, sustainable action in poor communities [6]. The Congregations Taking Action against NCDs (CONTACT) Study intervention aims to strengthen primary healthcare systems in the Caribbean using a novel approach that integrates places of worship into the primary care pathway. Its premise is that trained health advocates (HAs) from their local places of worship (POWs), supervised by nurses at the nearest primary healthcare centre (PHC), could increase access to primary care, particularly among those with distrust of health services and difficulties with access. HAs in churches, mandirs (Hindu temples) and mosques could be trained to communicate simple health promotion messages (e.g., importance of fruit and vegetable consumption), conduct screening procedures (e.g., measuring blood pressure using automated validated devices) and work collaboratively with nurses to ensure appropriate referrals of congregants and adherence to recommended management.

CONTACT builds on social capital theory [7, 8] and the growing evidence base relating to faith-based interventions [9–12] and community health workers (including community volunteers) in community-based interventions [13, 14]. Systematic reviews have shown positive

effects of faith-based interventions to promote health and wellness among vulnerable communities. For example, Sanusi et al (*PLOS Global Public Health*, 2023) concluded that "in addressing the global hypertension epidemic, cardiovascular health promotion roles of faith institutions probably hold unrealised potential in research and utility as viable assets or adjuncts to healthcare systems, crucially in low-income, religious or underserved communities" [15]. The moral status of places of worship within communities enhances acceptability of community-based interventions. This aligns closely with assets-based community approaches that emphasise the capabilities and assets of communities, rather than their deficiencies, to foster healthy living [16]. Places of worship are community health assets with extensive social networks that can be leveraged for health promotion and universal access to services. Attributes of community health assets include linkages with community-based civil society groups that provide a myriad of informal and formal support (e.g., counselling, food, healthcare, training, jobs) [17–19].

Like community health workers, HAs in POWs could be an enabling strategy for integrating community and health care systems and for providing preventive health services. Some of the benefits known for community health workers such as trusting relationships with patients, enhancing cultural relevance of health materials and information and cost-effective extensions of the health system could apply to HAs in congregations [13, 14]. As insiders of the different faith communities (Christians, Muslims and Hindus in the Caribbean), they may be able to shape the healthcare system to suit the needs of different populations, thus improving cultural accessibility of services, addressing health inequities, and strengthening health system performance and efficiency.

There is a general dearth of asset-based intervention studies within diverse religion groups, with most published reports focusing on Christian churches [20]. Additionally, most studies have focused on behavioural modifications at the individual level, delivered by external health professionals, rather than developing a systems-based framework or identifying constrainers and enablers for sustainable linkages at the church or community levels. By contrast, CONTACT uses a structured participatory approach [21] whereby policy actors, health centre practitioners and communities were able to jointly agree on a potential action plan for embedding places of worship in the primary care pathway via health advocates. An important aspect of that joint planning was ascertaining the level of readiness of relevant actors in each of the three countries for such a complex intervention. In this paper, we report on the first phase of the study, where we 1) explored what factors policy actors, healthcare practitioners, and civic society representatives felt could influence the ability of health centres to support the CONTACT intervention, then 2) used these findings to inform direct assessment of enablers and constraints at local PHCs (and POWs, reported separately).

## Design and methods

The first phase of the CONTACT Study involved mixed-methods, participatory approach to obtaining stakeholder input on the feasibility of the proposed intervention, and assessing the readiness of PHCs for integration of places of worship into the primary care system.

## Study setting: The Caribbean context

Three countries were selected for CONTACT due to their differing socio-economic, cultural and health system contexts (Table 1). Guyana is the third poorest country in the Western hemisphere with 55% of its citizens below the poverty line. There are low female labour force participation, excess suicide rates and high teenage pregnancy. Jamaica and Dominica follow closely behind Guyana in relation to Human Development Indices [22]. In Jamaica, rural

**Table 1. Economic and health indicators for Guyana, Jamaica and Dominica.**

| Characteristic | Guyana | Jamaica | Dominica |
|---|---|---|---|
| GDP per capita (USD) [27] | 9,374.8 | 4,586.7 | 7,560.0 |
| Main ethnic groups | Indian, African, Mixed, Indigenous | African | African, Kalinago (Indigenous) |
| Main religions | Christianity, Hinduism, Islam | Christianity | Christianity |
| % GDP allocated to health [28] | 4.9 | 6.1 | 5.5 |
| Estimated Indigenous population (%) | 10.5 [29] | <1% [30] | 2.9 [31] |
| Number of tertiary care hospitals | 1 | 8 | 1 |
| Number of physicians (/1,000 population) [32] | 1.4 | 0.5 | 1.1 |
| Number of nurses and midwives (/1,000 population) [32] | 3.5 | 0.9 | 6.1 |
| Hospital beds (per 1,000 people) [33] | 1.7 | 1.7 | 3.8 |
| Diabetes prevalence (% of population aged 20–79) [34] | 11.6 | 11.3 | 11.6 |
| Hypertension prevalence (% of population aged 30 to 79) [35] | 40.0 | 46.3 | 47.7 |
| Electronic information systems | no | embryonic | no |
| National surveillance of inequalities in health outcomes | no | no | no |
| % of population aged 65 years and above | 7 [36] | 9 [36] | 11.5 [31] |
| Poverty headcount ratio $1.90/day (%) [37] | 2.5 | 1.0 | N/A |
| Adult obesity(%) [37] | 20.2 | 24.7 | 27.9 |
| Death rate from NCDs (% adults aged 30–70 years) [37] | 30.5 | 14.7 | N/A |
| Healthy Life Expectancy at birth (years) [38] | 58.3 | 66.9 | N/A |
| Crude suicide rates (/100,000) [38] | 29.2 | 2.2 | N/A |

poverty and urban slums challenge equitable health care delivery. In Dominica, 29% of citizens live below the poverty line, with the highest proportion among the Indigenous Kalinago [23], upon whom we focus in this study. Whereas Guyanese have broad representation across Christianity, Hinduism and Islam, Jamaicans and the Kalinago are mainly Christians. Health systems in all three countries are marked by workforce shortages, concentration of health services in urban areas and inadequate integration between primary and secondary care. All three countries have decentralised, public sector-led health systems consisting of health regions and networks of health centres, with care provided free of cost at the point of access. Several initiatives in the Caribbean demonstrate the potential of places of worship in health promotion but these focus on infectious diseases and preventive behaviours at the individual level [24–26].

## Concept mapping: Obtaining stakeholder input

Right from the project's conception, our approach centred on inclusivity and participation for the co-development and evaluation of the CONTACT interventions. The concept mapping process marked the study's initial planning phase and involved diverse professional and lay stakeholders, many of whom were congregants themselves. Their combined perspectives informed all other phases of the study including the conduct of readiness assessments and design of survey instruments. Concept mapping is a mixed-methods approach using qualitative procedures to generate ideas on a topic across a group, formulate the research questions, followed by quantitative methods to synthesise and represent the group's ideas visually in a series of maps. Concept mapping phases were conducted as described by Trochim, and as illustrated in Fig 1 [39, 40].

In Guyana and Jamaica, data collection workshops were held at the participating universities, and in Dominica at a health centre in the Kalinago Territory.

*Step 1*: Purposeful sampling was used to recruit 48 stakeholders (Guyana = 18; Jamaica = 14; Dominica = 16) with broad multi-sectoral representation from Ministries of Health (n = 20),

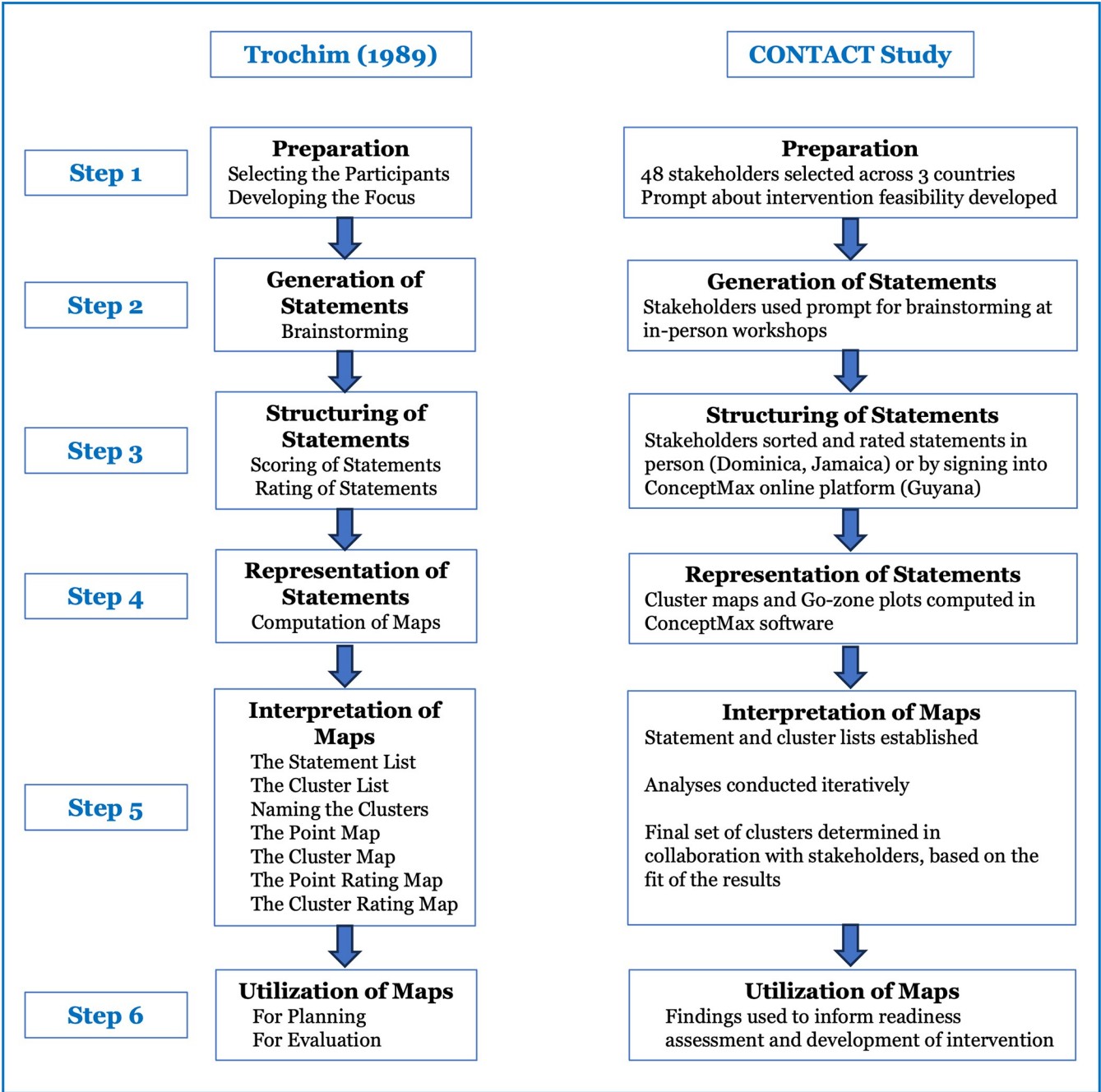

**Fig 1. Illustration of how the CONTACT study implemented Trochim's six-step concept mapping process.**

the religious community (n = 15), non-government organisations (n = 5), and academia (n = 8). S1 Table shows the sectoral composition of stakeholders by country. During the recruitment process, the ideas for intervention development were described to the stakeholders.

*Step 2*: Stakeholders then 'brainstormed' statements on a sheet of paper, guided by the focussed prompt *'Factors that will affect the ability of health centres to promote the intervention are . . .'*. Following brainstorming, the research team reviewed the statements and edited them for clarity and grammar, but not for content, and agreed the final list with stakeholders.

*Step 3*: Stakeholders sorted the statements into what they perceived to be conceptually similar categories. Next, each participant rated the perceived importance and feasibility (of change/ improvement) of each statement on 5-point Likert scales: 1 = 'not at all important' to 5 = 'very important' and 1 = 'not at all feasible' to 5 = 'very feasible', respectively.

*Step 4*: Analyses were conducted using the Concept Systems MAX software following established methods described in detail elsewhere [41, 42]. In summary, each sorting category was first converted to a 0, 1 co-occurrence matrix, that had as many rows and columns as there were statements. It contained 1 in a cell if the row and column statement pair were placed by the participant in the same category and a 0 if the statements were not sorted together. These matrices were then summed across all stakeholders, yielding a similarity matrix that indicated the number of stakeholders that sorted each pair of statements together. Multidimensional scaling analysis used (dis)similarity data and represented them as distances in Euclidean space. Each statement was assigned an 'x' and 'y' value from these analyses which were used in hierarchical cluster analysis to partition the statements into non-overlapping clusters. The statements closer to each other on the map were expected to have been sorted similarly across stakeholders. A bridging value was computed for each statement and the average for each cluster, which ranged from 0 to 1; the lower the value, the better the coherence from sorting across stakeholders. Ratings for importance and feasibility were then averaged for all participants and overlaid on the cluster concept maps. Bivariate 'Go-Zone' plots were then produced which compared plots of patterns of ratings (feasibility and importance) at the statement level. The bivariate space was divided into quadrants based on the average x and y values. For example, when comparing an importance and feasibility rating of the statements, the Go-Zone was the quadrant showing the statements simultaneously rated above average in both importance and feasibility. Go-Zones are particularly valuable for intervention development, and each statement could be described as what stakeholders perceived as an actionable priority.

*Step 5*: The analyses were conducted iteratively, and the final number of clusters were determined with stakeholder input, based on both the fit and interpretability of the results. This allowed synthesis of the input of multiple stakeholders into a unified quantitative representation (Go Zone) of factors that could influence the feasibility of the proposed intervention.

*Step 6*: The findings subsequently informed readiness assessments of the health centres and intervention development (reported separately).

## Assessment of capacity and readiness of health centres

Twelve health centres were purposefully sampled with the support of the Ministries of Health based on context-specific needs such as primary care staffing levels, NCD prevalence, size and geographical distribution of religious groups, poverty indices, and relative ease of access to community sites. As with other low- and middle-income countries (LMICs), health worker maldistribution exists, with higher urban concentrations compared with rural regions where large proportions of the population reside [43].

In Guyana, six health centres in rural West Demerara (Region #3) were selected based on high prevalence of obesity, diabetes, and levels of deprivation, and a lack of public health programmes targeting NCDs. In Jamaica, 4 health centres in poor areas, two urban (Olympic Gardens and Norman Gardens) and two rural (Riversdale and Yallahs) were selected. In Dominica, the only two health centres in the Kalinago Territory (which is mainly mountainous and forest) were selected. A questionnaire was developed that included questions based on the findings from concept mapping and from the WHO handbook for Monitoring the Building Blocks of Health Systems [44]. The questions from the latter were adapted to suit the context of a single health centre. For example, service delivery was assessed using health centre-

relevant WHO tracer items for general and NCD-specific services. The questionnaire was completed by the district medical officer or the nurse-in-charge at the health centre. Questionnaire data were summarised qualitatively (descriptively) for individual health centres then synthesized for each country, according to the components of the WHO Building Blocks, grouped under broader themes derived from concept mapping.

Ethics approvals for CONTACT were received from the Ministry of Public Health and its Ethical Review Committee (Protocol # 253) in Guyana, the University of the West Indies Ethics Committee and the Ministry of Health & Wellness Medicolegal Committee in Jamaica (Protocols # 197/15/16, 2016/26), and Ross University School of Medicine Institution Review Board (Protocols # 015–112222016) and Ministry of Health in Dominica. Approvals were also received from the Kalinago Council in Dominica. Participants provided written informed consent prior to study participation.

## Inclusivity in global research

Additional information regarding the ethical, cultural and scientific considerations specific to inclusivity in global research is included in the S1 Checklist.

## Results

### Stakeholders' Go-Zone for the factors that could affect the ability of primary health care centres to promote the intervention

Brainstorming resulted in 108 statements (33 for Guyana, 23 for Jamaica and 52 for Dominica). Five cluster solutions fitted the data well and yielded interpretable results for each country. S2 Table shows for each country the statements within clusters, again with ratings for importance and feasibility). Service Capacity and Resources were common clusters for all three countries; additionally, Training was common to Guyana and Dominica. Average bridging values were also lowest for these clusters, reflecting strong agreement across stakeholders in each country in the sorting of the statements into these clusters.

Figs 2–4 display the average ratings for feasibility and importance of all statements across the four quadrants of the bivariate plots, with the colours denoting the cluster for the statements. The upper right quadrant, the Go-Zone, shows the statements that stakeholders rated as most important and feasible for embedding places of worship into the primary care pathway for NCD prevention. Just over one-third of the statements were in the Go-Zone. The Go-Zone contained statements which largely reflected system level actions. Several were common across the three countries, but country specific actions were also evident.

The Go-Zone for Guyana contained statements mainly from clusters with strongest agreement across stakeholders (Service Capacity, Resources, Training). It included commitment of the Ministry of Health, leadership at the health centres, training, clarity of roles, communication, availability of congregants with skills to become health advocates and recognition of the value of the intervention by religious leaders. A notable feature of the bivariate plot was that there were just as many statements in the Go-Zone as in the lower right quadrant, which contains statements that were rated as relatively highly important but not highly feasible. Several statements were in the clusters 'Service Capacity' (e.g., lack of access to medicines and inability to manage referrals) and 'Resources' (e.g., understaffing, lack of trained staff to supervise health advocate). Another interesting feature related to the statements in the upper left quadrant (relatively feasible but not important). These reflected socio-cultural influences such as gender, ethnicity and cultural beliefs of health advocates.

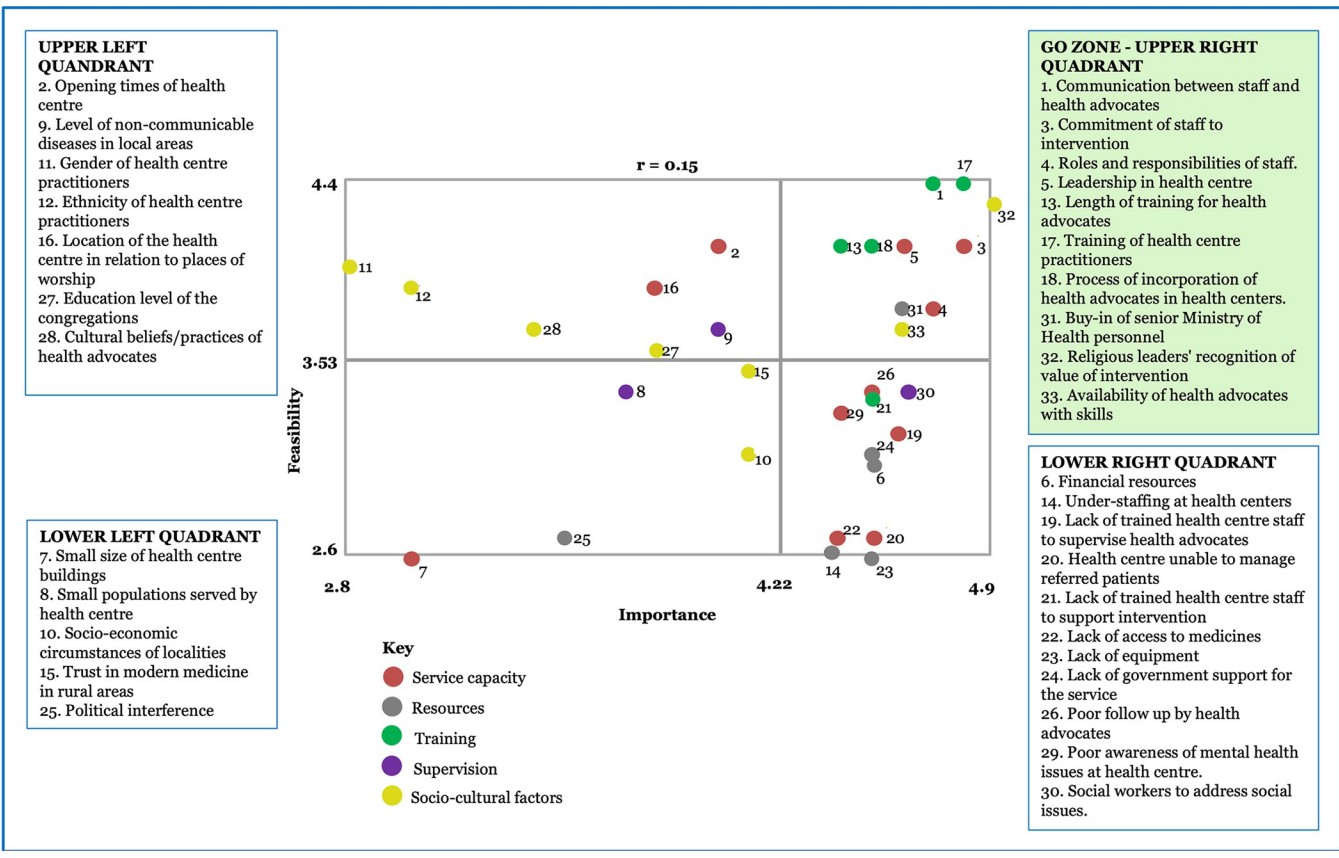

**UPPER LEFT QUANDRANT**
2. Opening times of health centre
9. Level of non-communicable diseases in local areas
11. Gender of health centre practitioners
12. Ethnicity of health centre practitioners
16. Location of the health centre in relation to places of worship
27. Education level of the congregations
28. Cultural beliefs/practices of health advocates

**GO ZONE - UPPER RIGHT QUADRANT**
1. Communication between staff and health advocates
3. Commitment of staff to intervention
4. Roles and responsibilities of staff.
5. Leadership in health centre
13. Length of training for health advocates
17. Training of health centre practitioners
18. Process of incorporation of health advocates in health centers.
31. Buy-in of senior Ministry of Health personnel
32. Religious leaders' recognition of value of intervention
33. Availability of health advocates with skills

**LOWER LEFT QUADRANT**
7. Small size of health centre buildings
8. Small populations served by health centre
10. Socio-economic circumstances of localities
15. Trust in modern medicine in rural areas
25. Political interference

**LOWER RIGHT QUADRANT**
6. Financial resources
14. Under-staffing at health centers
19. Lack of trained health centre staff to supervise health advocates
20. Health centre unable to manage referred patients
21. Lack of trained health centre staff to support intervention
22. Lack of access to medicines
23. Lack of equipment
24. Lack of government support for the service
26. Poor follow up by health advocates
29. Poor awareness of mental health issues at health centre.
30. Social workers to address social issues.

**Key**
- Service capacity
- Resources
- Training
- Supervision
- Socio-cultural factors

**Fig 2. Go-Zone map based on stakeholders' perspectives on feasible and important factors in response to the prompt: "Factors that will affect the ability of health centres to promote the intervention in Guyana are. . .".**

The Go-Zone for Jamaica contained statements from three clusters, with just under half from the two clusters that had strong stakeholder agreement, 'Resources' and 'Stakeholder Motivation', and the rest were from 'Health Advocate's role'. There were nuanced differences between the Go-Zones for Guyana and Jamaica. For example, statements relating to resources (financial, technological, human) were rated as relatively important and feasible in Jamaica, but in Guyana some of these were rated as relatively important but not feasible. Additional unique statements in Jamaica's Go-Zone cited individual responsibility for health among congregants, monitoring and feedback of progress and the inclusion of a focus on supporting those with chronic disease. As in Guyana, some service capacity issues (e.g., health centre nurses lack time to supervise health advocates) were rated as relatively important but not feasible.

The Go-Zone for Dominica also contained statements mainly from the three clusters with strongest agreement across stakeholders (Service Capacity, Resources, Training). There was more alignment with those for Guyana (e.g., communication, commitment of staff, training acceptability of the programme by the Ministry of Health) than for Jamaica. Unique statements rated relatively high for both feasibility and importance included enthusiasm for the proposed intervention, joint development of programme between churches and health centres, use of experts in the congregations, and attitudes of health centre staff to health advocates. A notable difference of the bivariate plot for Dominica was the central region where ratings for statements for feasibility and importance were not very low. These included understanding the

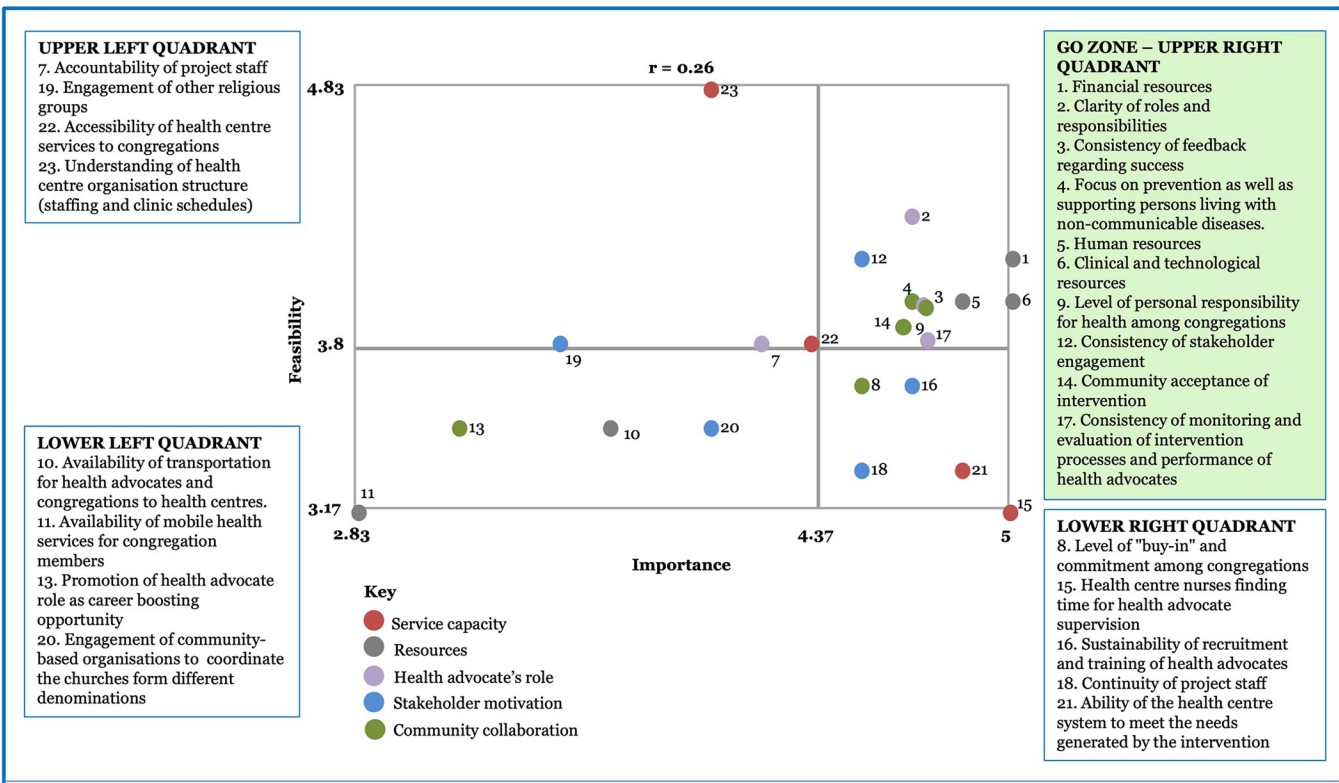

**Fig 3. Go-Zone map based on stakeholders' perspectives on feasible and important factors in response to the prompt: "Factors that will affect the ability of health centres to promote the intervention in Jamaica are…".**

history of the Kalinago, incentives for health workers, equipment for churches and overburdened staff.

## Assessment of readiness of health centres

Table 2 presents a summary of specific enablers and constrainers by country, drawn from the readiness assessments of 12 health centres. S3 Table gives detailed information for each health centre. Column 1 captures many of the issues raised in the concept maps, including service capacity, training of health workers, and resources which included access to technical tools and policy support for community interventions. Under-resourcing, lack of systematic training in NCDs and lack of policy support for embedding community interventions were common constrainers across the countries.

The six rural health centres in Guyana served between 3000–15000 people, had limited service capacity, but all had electricity and running water and were accessible by local public transport. Staff reported that there were monthly meetings with the regional health authorities to review needs, and all centres had a chronic disease clinic. An important enabler was that staff reported well-defined roles and responsibilities and appreciated the potential benefits of the CONTACT intervention. Variable levels of community engagement put some health centres in a better position to form linkages with places of worship in the community. Some of the contextual constrainers aligned with what was reported as relatively important but not feasible in the concept map. These included a lack of essential medicines and basic equipment; limited knowledge and awareness of NCD protocols and health promotion practices; reliance of referrals for specialist care on services offered at the regional or central referral hospital; understaffing; lack of regular training of staff in NCD prevention and management; and lack of

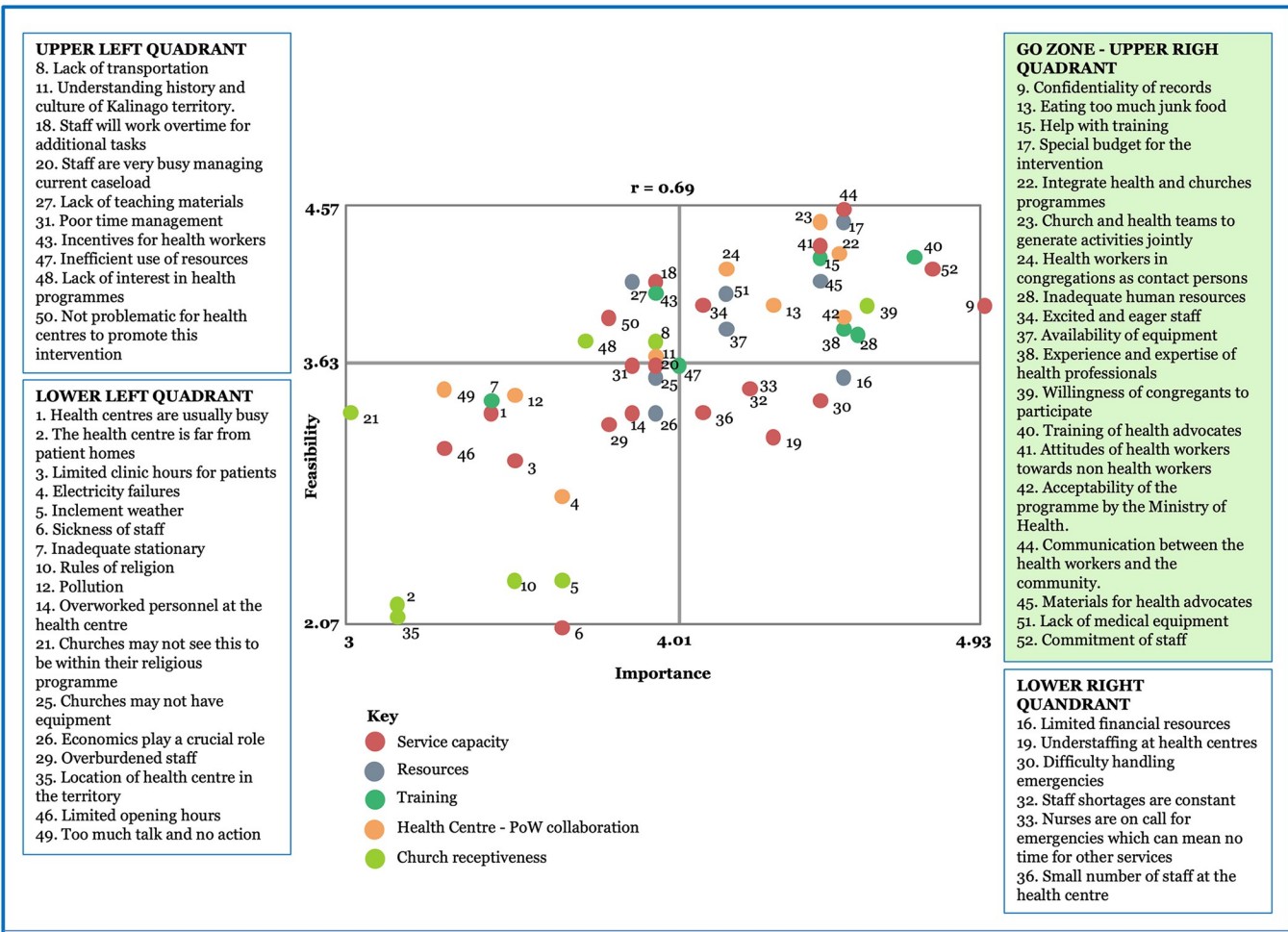

**UPPER LEFT QUADRANT**
8. Lack of transportation
11. Understanding history and culture of Kalinago territory.
18. Staff will work overtime for additional tasks
20. Staff are very busy managing current caseload
27. Lack of teaching materials
31. Poor time management
43. Incentives for health workers
47. Inefficient use of resources
48. Lack of interest in health programmes
50. Not problematic for health centres to promote this intervention

**LOWER LEFT QUADRANT**
1. Health centres are usually busy
2. The health centre is far from patient homes
3. Limited clinic hours for patients
4. Electricity failures
5. Inclement weather
6. Sickness of staff
7. Inadequate stationary
10. Rules of religion
12. Pollution
14. Overworked personnel at the health centre
21. Churches may not see this to be within their religious programme
25. Churches may not have equipment
26. Economics play a crucial role
29. Overburdened staff
35. Location of health centre in the territory
46. Limited opening hours
49. Too much talk and no action

**GO ZONE - UPPER RIGH QUADRANT**
9. Confidentiality of records
13. Eating too much junk food
15. Help with training
17. Special budget for the intervention
22. Integrate health and churches programmes
23. Church and health teams to generate activities jointly
24. Health workers in congregations as contact persons
28. Inadequate human resources
34. Excited and eager staff
37. Availability of equipment
38. Experience and expertise of health professionals
39. Willingness of congregants to participate
40. Training of health advocates
41. Attitudes of health workers towards non health workers
42. Acceptability of the programme by the Ministry of Health.
44. Communication between the health workers and the community.
45. Materials for health advocates
51. Lack of medical equipment
52. Commitment of staff

**LOWER RIGHT QUANDRANT**
16. Limited financial resources
19. Understaffing at health centres
30. Difficulty handling emergencies
32. Staff shortages are constant
33. Nurses are on call for emergencies which can mean no time for other services
36. Small number of staff at the health centre

**Key**
- Service capacity
- Resources
- Training
- Health Centre - PoW collaboration
- Church receptiveness

**Fig 4. Go-Zone map based on stakeholders' perspectives on feasible and important factors in response to the prompt: "Factors that will affect the ability of health centres to promote the intervention in Dominica are. . .".**

government support for developing systemic partnerships with community structures. In addition, staff were placed by the Ministry of Health and did not necessarily live in the area served by the health centres, which hindered community engagement activities.

In Jamaica, the two rural health centres served 39,000 and 15,000 each while urban health centres served 25,000 and 15,000 each. There were several enablers specific to Jamaica. Health centres were managed by a Public Health Nurse, offered NCD-related services including diabetes, chronic disease and mental health clinics, had monthly sessions held by health educators from the regional office, though this could cover any topic, and all had access to an onsite or nearby community pharmacy. Patients could be referred for specialist care at the Kingston Public Hospital or University hospital. Mental health officers assisted with referrals for social care. Nurses received some training in NCD prevention and management during monthly review meetings at the Regional Authority. Although willing to support CONTACT, there were concerns about supervision of health advocates incurring additional workload. All health centres were understaffed with crowded waiting areas. As in Guyana, staff were centrally placed in health centres and many did not live in the area they served, community health workers were not trained in NCD prevention or management and there was limited knowledge of the NCD guidelines, especially in the rural health centres. Government support for

**Table 2.** Readiness assessments of health centres in Guyana, Jamaica and Dominica: Summary of enablers (blue cells) and constrainers (orange cells) to promote the embedding of places of worship into the primary care pathway.

| COUNTRY/ DOMAIN ASSESSED | GUYANA (6 Healthcare Centres serving~ 45,000) | JAMAICA (4 Healthcare Centres serving ~ 94,000) | DOMINICA (2 Healthcare Centres serving ~ 2–3,000) |
|---|---|---|---|
| **SERVICE** | | | |
| **General infrastructure, amenities, accessibility** | All sites functional, 3 very small. Accessible via public or private transport | All small and functional, accessible by public transportation. Telephone and water available | All functional and opened daily, with nurse on call 24hrs for emergencies. Water and electricity available |
| | Variable access to electricity, water and telephone. No internet. Limited space to accommodate increased patient flow in 3 sites. Open weekdays only. | Inadequate for current case load and therefore for any increase in patient flow in 3 sites. No internet. | No internet or landline. Limited space and storage in 1 site. Not easily accessible by public transportation. |
| **Governance/supervision arrangements** | Ministry of Health and Regional Democratic Council employ staff. Senior Health Visitor supervises nurses. Nurse in charge manages health centre and supervises nurses and community health workers. Doctors assigned to health centres. | Regional Health Authority employs staff. Public Health Nurse manages health centre and supervise midwives, who supervise community health workers. Senior Public Health Nurse for each health district. Nutritionist and mental health worker assigned to clinic. | Ministry of Health employs staff. One health centre built and owned by Kalinago Council and leased by Ministry of Health. Community Health Nurse, employed at the health district level, supervises all nurses in the district. |
| **Services offered** | Bi/weekly NCD services including diabetes clinic, screening and health promotion activities. | Weekly NCD clinics including diabetes. Nutrition and mental health clinics 1–3 times per month. | Daily NCD services including screening and health promotion. Quarterly diabetes clinic, monthly hypertensive clinic. Specialist mental health clinics on request. |
| | Mental health clinic in one site only. | | |
| **Essential medicines** | Variable with a history of drug shortfall for common chronic diseases. | Pharmacies on 2 sites provide most routinely prescribed drugs. 2 have access to a community pharmacy. | Variable with small quantities for some conditions or not supplied by government. |
| **Equipment/tools to aid front line delivery** | Variable access to basic equipment. Blood samples not taken on site. | Variable access to basic equipment. Blood samples taken twice per month and analysed elsewhere. | Access to basic equipment. Blood samples taken twice per month and analysed elsewhere. |
| **Health information systems** | Paper-based, well organised in most sites. | Paper based dockets managed by the records clerk. | Paper-based; NCD register well-maintained. |
| **Implementation of NCD protocols** | Three sites aware of NCD protocols, partially implemented. | NCD protocols available, partially implemented. | National hypertension and diabetes protocol implemented. |
| **Referral pathways to specialist care** | Doctors refer patients to the District/Regional/Central Referral Hospitals. | District Medical Officer refers to specialist or tertiary care services. | District Medical Officer refers to main hospital or private specialist. |
| **Referral pathways to social care** | Doctors refer to the Regional hospital's social worker | Mental health officer assists with social care, community health aides make referrals at 1 site | Referral done by Community Health Nurse for welfare/social services and social support programmes |
| | | Little or no support for referrals | |
| **Regular monitoring and evaluation reports** | Regular paper-based reporting. Nurses report monthly to Senior Health Visitor. Surveillance data sent monthly to the Surveillance Unit of the Ministry of Health | Regular paper-based reporting. Public Health Nurse updates surveillance data, which is sent monthly to the Parish health department | Regular paper-based reporting. Surveillance data reported to the Family Nurse Practitioner and Community Health Nurse monthly/quarterly for discussion at health centre meetings |
| **HEALTH WORKERS** | | | |
| **Sufficient staff to manage patient flows** | Inadequate staff to patient ratio | Inadequate staff to patient ratio | Inadequate staff to patient ratio |
| **Training of health centre staff in NCD prevention and control** | Most staff received some training in NCD management | Monthly in-service training for nurses by the regional authority | Monthly in-service training for nurses by practitioners. Ad hoc workshops by the Pan American health Organization/Ministry of Health |
| | No regular refresher training | Community health workers are not trained in NCD | |
| **Training of health centre staff in communication** | No regularly scheduled training | No regularly scheduled training | No regularly scheduled training |
| **Willingness and motivation to support CONTACT** | Appreciated the potential benefits of CONTACT | Appreciated the potential benefits of CONTACT | Appreciated the potential benefits of CONTACT |
| **Clarity of roles and responsibilities of all practitioners** | Staff aware of their roles and responsibilities in most sites | Staff aware of roles and responsibilities | Staff aware of roles and responsibilities |

*(Continued)*

**Table 2.** (Continued)

| COUNTRY/ DOMAIN ASSESSED | GUYANA (6 Healthcare Centres serving~ 45,000) | JAMAICA (4 Healthcare Centres serving ~ 94,000) | DOMINICA (2 Healthcare Centres serving ~ 2–3,000) |
|---|---|---|---|
| **Health Centre managers have the autonomy to support CONTACT** | Approval needed from Ministry of Health and Regional Health Officer | Approval needed from the Regional Health Authority | Approvals needed from Ministry of Health and Kalinago Council |
| **COMMUNITY COLLABORATIONS** | | | |
| **Community engagement/health promotion** | Little collaboration with places of worship | Limited community engagement | Numerous, including schools and churches |
| **Provision of ongoing care of patients with chronic disease in the community** | Community health workers provide limited services in the community | Community health workers provide limited services in the community | Home based care by nurse, community health aide |
| **GOVERNMENT SUPPORT** | | | |
| **Support for integrating community programmes into primary care pathway, including financing mechanisms** | Limited support for community interventions. | National Health fund provides sporadic activities in collaboration with non-government organisations<br><br>Limited support for community interventions. | Limited support for community interventions. |
| **Support for use of locally relevant tools, information systems and guidelines** | Inadequate, with staff creating health posters alongside those provided by the Ministry. No diagnostic tools (e.g. risk scores) | Inadequate with no reports of protocols or diagnostic tools such as risk scores. | Some protocols available from the Ministry of Health and Community Health Nurse supports implementation. |
| **Support for local decision making** | Approval for interventions made at the regional level if no budgetary implications. | Approval for interventions made at the regional level | Approval for interventions made at the regional level |

community interventions were limited to ad hoc health fairs or health promotion in schools by the health educator, with no known established health centre-community interface.

The health centres in Dominica were small and served about 2000 people. The governance is underpinned by historic Kalinago-Government arrangements. The building for one of the health centres was owned by the Kalinago People and leased to the Ministry of Health, whose responsibility was the planning and provision of health care. The centres were well organised with regular NCD and chronic disease clinics that included preventative services, such as hypertension screening and regular health promotion sessions in the waiting areas of the health centres and also in the community. Unlike the other two countries, there was an NCD register with details of diagnosis and referrals for specialist care. Nurses had monthly in-service training on NCD prevention and management, were enthusiastic about the intervention and felt they had some autonomy to implement the intervention. There were several examples of health promotion activities in the community such as lectures by health centre staff in different community settings, but these targeted individual behaviours rather than formal partnership working with community systems. Some of the key constrainers included poor accessibility due to lack of public transport in the Kalinago Territory with nurses and community health workers walking hours over hilly terrain to provide home-based care, understaffing with bi-monthly visits by a District Medical Officer, nurses shared across the two centres and being on call for 24 hours, with emergencies prioritised over regular services. There was also a lack of access to sufficient essential medicines and to on-site laboratory testing. Government support was variable with delays in approvals and inadequate financing that targeted community-based prevention initiatives.

## Discussion

Grounded in participatory and systems thinking perspectives, we consulted civic leaders, policy actors and practitioners regarding readiness of health centres in Guyana, Jamaica and

Dominica. Co-developed concept maps illustrated optimism for the CONTACT intervention and that service capacity, resources and training could influence implementation across all countries, with specific feasible and important actions relating to the country context. Readiness assessments of local primary care centres revealed a willingness of staff to support the intervention. Under-resourcing including under-staffing and inadequate essential NCD medicines, lack of systematic training in NCDs and of policy and technical support for integrating community interventions were common constrainers. Despite their historic context of socio-political disadvantage and constrained capabilities, the most enabling context for embedding churches into the primary care pathway appeared to be with the Indigenous Kalinago in Dominica. Structural and policy support were identified as critical for creating place of worship-primary health care partnerships.

A unique finding is the ease with which a diverse set of professional and lay stakeholders engaged with thinking about how to connect two complex systems—places of worship and primary health care centres. Implicit in the concept mapping process was the integration of their knowledge and beliefs about these systems. Attendance at places of worship is high in all three countries so stakeholders who participated in concept mapping were also invariably congregants (e.g., doctors, nurses, government officials); clearly that would have influenced their perceptions about the viability of places of worship and health centres working in partnership. The optimism of the Go-Zones, however, is still remarkable given the significant challenges from under-resourcing reported by health centre staff in the readiness assessments. The availability of trained health workers in NCD prevention and control, essential medicines, technological resources and information systems are all known to be critical gaps in the readiness of primary care in LMIC. With low staffing levels reported across the three countries, additional health workers may be required to supervise health advocates and manage the potential initial increase in newly identified cases at the health centres from referrals by health advocates. Nurses received regular training in NCDs in Jamaica and Dominica, but community health workers, whose primary responsibility is to be the liaison between the health centres and the communities, were not trained in NCDs. Given that of all health centre staff, only community health workers were generally from the communities served by the health centres in Guyana and Jamaica, the training of both community health workers and nurses could be critical to support CONTACT. Inadequate access to essential NCD medicines and to specialist services, particularly in Guyana and Dominica, raises ethical concerns of identifying new cases knowing there is no pharmacological treatment option and could have negative impact on health advocates' motivation and performance. Poor provision of onsite diagnostic tools and paper-based information systems could also hinder management of referrals by health advocates and follow-up in the community and through the care pathway.

At the global level, the NCD agenda set by transnational agencies such as the WHO and the United Nations recognises the value of population-based health services and of integrating health and social care for home-based and community care. A primary health care-place of worship collaborative model has the potential to be a poverty reduction strategy by engaging with the social determinants of health, reduce morbidity and mortality, increase productivity and quality of life [45, 46]. Across all countries, government support was felt to be inadequate for the creation of systemic health centre-community systems interface. Decision-making was devolved to regional/district level for the organisation and management of care, but health centre staff reported a lack of autonomy or capacity to initiate innovative systemic changes due to a lack of financing, training and governance arrangements. Rehabilitation and long-term care at home were limited in all three countries, but even where they existed, collaborative working between health centres with these services was limited, with a general lack of systematic referral systems from primary care to social care.

The political determinants of primary care have seldom been reported but cogent arguments have been made about the disproportionate emphasis on short-term gains impeding policy integration (e.g. health, welfare, education). Such integration addresses the interconnected causes of health inequities in the economies of middle income countries transition [4, 47]. Governments in the region have pledged commitment to several NCD Strategic Plans over the last decade but our findings show inadequate implementation. For example, the region-wide Pan American Health Organisation (WHO) Strategic Plan of Action for the Prevention and Control of NCDs in countries of the Caribbean Community (2011–2015) emphasised risk factor reduction and health promotion, patient self-management education, surveillance, monitoring and evaluation, public policy, advocacy and communication [48]. The Guyana NCD Strategic Plan (2013–2020) highlights the importance of promoting health at the population and community level [49]. The Jamaica NCD Action Plan recognises competence of the health workforce as a key issue when tackling NCD burden and lists it as a core principle guiding their plan [50]. The Dominica Strategic Plan for Health [51] emphasised the importance of building alliances and promoting multisectoral partnerships in NCD prevention and control.

Notwithstanding these challenges, the findings revealed notable levers that could support a feasibility study of CONTACT. The following highlights some of these enablers. Policy actors at national and community levels and religious leaders who participated in concept mapping are potential champions for CONTACT. This will be of paramount importance to enable supportive decision making in the policy and community spaces. An important aspect of readiness was the availability of community health workers in all three countries, and public health nurses in Jamaica. Their roles currently focus heavily on maternal and child health and infectious diseases but their knowledge of the communities and willingness to support CONTACT on the frontline will be critical. Given the high level of attendance to places of worship, there is the potential for health workers in the congregations to train health advocates, and to nurture community support which is essential for the credibility of health advocates.

The Pan American Health Organisation (PAHO) provided technical support for health centre staff in all three countries and could support the development and maintenance of relevant NCD training programmes for health advocates. PAHO provides not only technical help but financial support to the Ministries of Health. Co-development of CONTACT with PAHO is critical for strategic advocacy in government circles particularly for financial and governance support. In the Kalinago context, a participatory approach to systems thinking and safeguarding the health and well-being of their community aligned intuitively. Longstanding Kalinago activism has driven recognition of their rights, although complex issues persist in relation to decision-making between the Kalinago Council and the Ministry of Kalinago Affairs, land security, discrimination and socio-economic development of their Territory. Despite the severe national disruption from Hurricane Maria soon after concept mapping, the Kalinago remained positively engaged with the idea of CONTACT and felt they could advocate for the system level changes needed.

A major strength of this study is the use of concept mapping, an innovative, participant-driven methodology. It allowed, and allows generally, stakeholders from different sectors and with different levels of influence to share their opinions about important and achievable actions, and to shed light on the enablers/constraints [39]. The concordant themes with nuanced differences in specific actions across the countries emerged despite the fact that group concept mapping took place separately in the three countries. To date, faith-based interventions have largely focused on individual level behavioural modification; our findings signal an appetite for a systems-based framework tailored to local context and with equal partnerships.

While stakeholder's Go-Zones enabled the formulation of a plan, substantive data from the readiness assessments provided important contextual details related to training, resources,

governance and support to guide the development of a plan for feasibility testing of CON-TACT. For example, concerns were raised about the availability/capacity of congregants to be health advocates and implications for workload of supervisory nurses. In a forthcoming paper, we will report on the second development phase which focused on issues of recruiting and appropriate training of health advocates. There were potential limitations. We caution against the generalisability of concept maps given the selected motivated stakeholders; further, the process ran over many days and stakeholder burden could have been constraining. Despite the ethos of a participatory approach, the dynamics of group diversity could have prevented some from fully engaging. Twelve health centres were chosen on government's advice on needs of communities. The infra-structural inadequacies suggest that choices of health centres were not politically motivated, but they should not be regarded as generalisable to all health centres in the country.

## Supporting information

**S1 Checklist. Inclusivity in global research.**
(DOCX)

**S1 Table. Composition of stakeholders by country.**
(PDF)

**S2 Table. Themes; bridging values; average importance and feasibility ratings and statements for stakeholders based on their response to the prompt** *'Factors that will affect the ability of health centres to promote this intervention are. . .".*
(PDF)

**S3 Table. Readiness assessments of individual health centres: Details of enablers and barriers to promoting the embedding of places of worship into the primary care pathway.**
(PDF)

## Author Contributions

**Conceptualization:** Reeta Gobin, Robert Nasiiro, Seeromanie Harding.

**Data curation:** Davon Van-Veen.

**Formal analysis:** Davon Van-Veen.

**Funding acquisition:** Seeromanie Harding.

**Investigation:** Reeta Gobin, Troy Thomas, Sharlene Goberdhan, Manoj Sharma, Robert Nasiiro, Rosana Emmanuel, Madan Rambaran, Shelly McFarlane, Christelle Elia, Davon Van-Veen, Tiffany Palmer, J. Kennedy Cruickshank, T. Alafia Samuels, Rainford Wilks, Seeromanie Harding.

**Methodology:** Reeta Gobin, Troy Thomas, Robert Nasiiro, Madan Rambaran, Shelly McFarlane, Ishtar Govia, Ursula Read, J. Kennedy Cruickshank, T. Alafia Samuels, Rainford Wilks, Seeromanie Harding.

**Project administration:** Reeta Gobin, Seeromanie Harding.

**Supervision:** Reeta Gobin, Robert Nasiiro, Ishtar Govia, Ursula Read.

**Visualization:** Christelle Elia.

**Writing – original draft:** Reeta Gobin, Sharlene Goberdhan, Seeromanie Harding.

**Writing – review & editing:** Reeta Gobin, Troy Thomas, Sharlene Goberdhan, Manoj Sharma, Robert Nasiiro, Rosana Emmanuel, Madan Rambaran, Shelly McFarlane, Christelle Elia, Ishtar Govia, Tiffany Palmer, Ursula Read, J. Kennedy Cruickshank, T. Alafia Samuels, Rainford Wilks, Seeromanie Harding.

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
