## [Decision Letter · Decision Letter 0]

21 Feb 2023

PONE-D-22-23088Readiness for embedding places of worship into the primary care pathway for prevention and control of noncommunicable diseases: a participatory mixed method study in the CaribbeanPLOS ONE

Dear Dr. Goberdhan,

Thank you for submitting your manuscript to PLOS ONE. After careful consideration, we feel that it has merit but does not fully meet PLOS ONE’s publication criteria as it currently stands. Therefore, we invite you to submit a revised version of the manuscript that addresses the points raised during the review process.

We look forward to receiving your revised manuscript.

Kind regards,

Pracheth Raghuveer, MD, DNB

Academic Editor

PLOS ONE

Journal Requirements:

Reviewers' comments:

Reviewer's Responses to Questions

**Comments to the Author**

1. Is the manuscript technically sound, and do the data support the conclusions?

Reviewer #1: Partly

2. Has the statistical analysis been performed appropriately and rigorously? 

Reviewer #1: N/A

3. Have the authors made all data underlying the findings in their manuscript fully available?

Reviewer #1: Yes

4. Is the manuscript presented in an intelligible fashion and written in standard English?

Reviewer #1: Yes

5. Review Comments to the Author

Reviewer #1: Thank you very much for allowing me to review this great piece. Considering the lack of coherence, poor methodology and results presentation, I suggest a substantial revision of this paper.

Introduction: Well-written introduction; however, few observations that may improve the quality-

-Line 85- Systematic reviews have shown positive impacts of interventions---on what? Health reported outcomes, self-management practices, screening,….( please cite evidence). Furthermore line 89 there full stop should be after reference 13,14.

- The background is too long, and I would authors to make it succinct ( no more than 1200 words). A strong rationale need to be presented for the participatory and system thinking approach. Please take a look at the paper by Yadav et.al, who have presented a strong rationale for engaging stakeholders in the design process and you may pull something from this paper.

Yadav, U.N., Lloyd, J., Baral, K.P. et al. Using a co-design process to develop an integrated model of care for delivering self-management intervention to multi-morbid COPD people in rural Nepal. Health Res Policy Sys 19, 17 (2021). https://doi.org/10.1186/s12961-020-00664-z

-Formatting issues are throughout the paper and suggested to follow journal guidelines.

_ there is a lack of coherence between the paragraphs in the introduction. The last paragraph of the Caribbean context looks out of sorts for me. You need to have a strong rationale and then connect your objectives with it. To me, introduction looks very messy.

Design and methods section:

- First you need to explain your methodology and then you have to cover following: study setting, study population, stakeholders engagement- explain concept mapping( steps 1 to six should be presented in the table. More details are required on data sources and analysis plan with supporting theory. Did authors applied any theory to analyse the data.

Assessment of capacity and readiness of health centres: The selection of health centres can be presented in diagrammatic form rather than explaining.

Line 228- A questionnaire was developed –not created

-Details regarding the development of intervention are missing.

Results: The title says- A participatory mixed method but it’s unclear which mixed method design did authors used.

I can see two convoluted themes for this massive study which is not reader-friendly at all. Authors should present the results in such a way that readers will understand them in an easy way. I am getting anywhere while going through this paper as it’s too lengthy and does not have any coherence. I would suggest authors apply a clear theory to present their findings; based on which you can refine your discussion.

Please use more infographics which may reduce your word count.

The author’s team must work substantially to establish story coherence and take home the message for readers. Overall, this paper is too long, which may not attract readers.

6. PLOS authors have the option to publish the peer review history of their article (what does this mean?). If published, this will include your full peer review and any attached files.

Reviewer #1: No

---

## [Author Response · Author response to Decision Letter 0]

30 Aug 2023

Responses to Reviewer (Reviewer #1) Comments to the Author

[Note to Editor: Please be advised that all updated line numbers refer to the clean “Manuscript” file, not the file with tracked changes, as having tracking on can affect line numbers.]

General

1. Comment: Thank you very much for allowing me to review this great piece. Considering the lack of coherence, poor methodology and results presentation, I suggest a substantial revision of this paper.

Response: We wish to thank the reviewer for their thorough consideration and detailed comments. We hope that they have now been satisfactorily addressed, and that the manuscript is even stronger.

Introduction

2. Comment: Well-written introduction; however, few observations that may improve the quality-

-Line 85- Systematic reviews have shown positive impacts of interventions---on what? Health reported outcomes, self-management practices, screening,….( please cite evidence).

Response: In the original manuscript we stated that systematic reviews showed positive effects of faith-based interventions to “promote health and wellness among vulnerable communities”. We have now added an example for increased clarity. (Lines 89-92) 

3. Comment: Furthermore line 89 there full stop should be after reference 13,14. 

Response: The full stop in this sentence was before the references in keeping with the rest of references (and referencing style used) in the paper. This is now line 87.

4. Comment: The background is too long, and I would authors to make it succinct (no more than 1200 words). 

Response: The Introduction in the original submission was ~1155 words, i.e., already <1200 (not counting the words in Table 1). However, having moved the paragraph on Caribbean Context and Table 1 into the Methods section and revised the rest of the Introduction, the word count is now 761. 

5. A strong rationale need to be presented for the participatory and system thinking approach. Please take a look at the paper by Yadav et.al, who have presented a strong rationale for engaging stakeholders in the design process and you may pull something from this paper.

Yadav, U.N., Lloyd, J., Baral, K.P. et al. Using a co-design process to develop an integrated model of care for delivering self-management intervention to multi-morbid COPD people in rural Nepal. Health Res Policy Sys 19, 17 (2021). https://doi.org/10.1186/s12961-020-00664-z

Response: Our original manuscript included numerous references [7-14] related to social capital theory and systems thinking; however, we have now added the Yadav reference to strengthen our examples of participatory approaches (Ref 21, Line 117).

6. Formatting issues are throughout the paper and suggested to follow journal guidelines.

Response: We have thoroughly reviewed the updated manuscript to ensure that all formatting guidelines have been followed.

7. there is a lack of coherence between the paragraphs in the introduction. The last paragraph of the Caribbean context looks out of sorts for me. You need to have a strong rationale and then connect your objectives with it. To me, introduction looks very messy. 

Response: we believe that the first paragraph of our introduction provides a strong rationale for our study, while paragraph 2 and the updated paragraph 5 (lines 116 to 121) support our objectives of obtaining stakeholder input and reviewing PHC readiness. To improve coherence, we have moved the paragraph on Caribbean Context and Table 1 into the Methods, under a new subheading “Study Setting”.

Design and methods section

8. First you need to explain your methodology and then you have to cover following: study setting, study population, stakeholders engagement - explain concept mapping (steps 1 to six should be presented in the table.

Response: We have revised the Methods section to include a brief introductory statement about the methods used (lines 131-133), a new subsection on Study Setting (information on the Caribbean Context from the Introduction) (lines 135-150, Table 1), and a new figure illustrating the recommended steps for concept mapping and how these were applied in the CONTACT study (caption at Line 167). 

9. More details are required on data sources and analysis plan with supporting theory. Did authors applied any theory to analyse the data. 

Response: Steps 1 to 3 (Lines 172-184) describe our data sources and data collection process for the concept mapping aspect, while data sources for the PHC readiness assessment are described in the first two paragraphs of this subsection (Lines 215-235). S1 Table provides further information on the background of the stakeholders who participated in the concept mapping, S2 Table presents the actual statements made by the participants, and S3 Table shows the detailed assessment of each health centre across the three countries. 

Regarding analyses, steps 4 and 5 (Lines 185-210) of the concept mapping process describe how stakeholders’ input was synthesized to provide a quantitative ranking of factors that they felt would influence feasibility of the CONTACT intervention. The following sentence was added to Step 5 to make this clearer “This allowed synthesis of the input of multiple stakeholders into a unified quantitative representation (Go Zone) of factors that could influence the feasibility of the proposed intervention.” The following sentence was added to the description of the PHC readiness questionnaire to clarify how the data were analysed: “Questionnaire data were summarised qualitatively (descriptively) for individual health centres then synthesized for each country, according to the components of the WHO Building Blocks, grouped under broader themes derived from Concept Mapping” (Lines 233-235).

10. Assessment of capacity and readiness of health centres: The selection of health centres can be presented in diagrammatic form rather than explaining. 

Response: The description of health centre selection (Lines 215-227) was brief (145 words), so we did not think a figure/diagram was necessary. 

11. Line 228- A questionnaire was developed –not created. 

Response: The word “created” was replaced with the word “developed” (this is now Line 227).

12. Details regarding the development of intervention are missing.

Response: At the time the concept mapping and PHC readiness were done, the intervention was not yet developed (apart from the broad concept of integrating health advocates at places of worship into the primary care system). Further, development required several other steps that could not be reported in this paper (including a place of worship assessment and a quantitative health survey of congregants). Therefore, it was not possible to provide details on the development of the intervention, beyond those included in the introduction (Lines 73-110). 

Results

13. The title says- A participatory mixed method but it’s unclear which mixed method design did authors used.

Response: As stated in Lines 173-176 of the original manuscript (now Lines 161-165), concept mapping is a mixed-methods approach that allows quantitative synthesis and representation of qualitative data. 

14. I can see two convoluted themes for this massive study which is not reader-friendly at all. Authors should present the results in such a way that readers will understand them in an easy way. I am getting anywhere while going through this paper as it’s too lengthy and does not have any coherence. I would suggest authors apply a clear theory to present their findings; based on which you can refine your discussion. 

Response: This phase of the CONTACT Study was primarily exploratory and descriptive i.e., identifying and presenting stakeholder perceptions about factors that would influence PHCs’ ability to participate in the proposed intervention, and using this information to help further assess the readiness of PHCs for such an intervention. As such, there was one underlying theme (PHC readiness) not two, and no theory or hypothesis was being tested. We did use more robust methods for analysing descriptive data than simple summary indices, which may have led to a high word count. We hope that the amendments described above have helped to improve the clarity and coherence of the paper.

15. Please use more infographics which may reduce your word count. 

Response: A figure summarizing the steps of concept mapping (now Figure 1) was added to improve clarity for persons unfamiliar with this approach; however we retained the narrative description for the same reason. We would be willing to place most of the detail in concept mapping Step 4 in Supporting Material if recommended by the Editor(s).

16. The author’s team must work substantially to establish story coherence and take home the message for readers. Overall, this paper is too long, which may not attract readers.

Response: We hope that the amendments described above have improved overall clarity and coherence of the paper. With regard to length, while we tried to write as concisely as possible, we thought it important to include sufficient detail given the complexity of the study and the novelty of our methods. The final word count of the paper is 4,911 (not including abstract, tables or references). 

Other Changes

As we were updating the manuscript, we took the opportunity to make the following changes:

1. Change the title from “Readiness for embedding places of worship into the primary care pathway for prevention and control of noncommunicable diseases: a participatory mixed method study in the Caribbean”, to “Readiness of primary care centres for a community-based intervention to prevent and control noncommunicable diseases in the Caribbean: a participatory, mixed-methods study”. The new title is a more accurate reflection of the focus of the paper.

2. Add an author who was inadvertently omitted from the first submission: Mr Davon Van-Veen, University of Guyana.

3. Update the Abstract.

---

## [Decision Letter · Decision Letter 1]

11 Jan 2024

PONE-D-22-23088R1Readiness of primary care centres for a community-based intervention to prevent and control noncommunicable diseases in the Caribbean: a participatory, mixed-methods studyPLOS ONE

Dear Dr. Goberdhan,

Thank you for submitting your manuscript to PLOS ONE. After careful consideration, we feel that it has merit but does not fully meet PLOS ONE’s publication criteria as it currently stands. Therefore, we invite you to submit a revised version of the manuscript that addresses the points raised during the review process.

Please submit your revised manuscript by Feb 25 2024 11:59PM. If you will need more time than this to complete your revisions, please reply to this message or contact the journal office at plosone@plos.org. Please include the following items when submitting your revised manuscript:A rebuttal letter that responds to each point raised by the academic editor and reviewer(s). You should upload this letter as a separate file labeled 'Response to Reviewers'.A marked-up copy of your manuscript that highlights changes made to the original version. You should upload this as a separate file labeled 'Revised Manuscript with Track Changes'.An unmarked version of your revised paper without tracked changes. You should upload this as a separate file labeled 'Manuscript'.If applicable, we recommend that you deposit your laboratory protocols in protocols.io to enhance the reproducibility of your results. Protocols.io assigns your protocol its own identifier (DOI) so that it can be cited independently in the future. For instructions see: https://journals.plos.org/plosone/s/submission-guidelines#loc-laboratory-protocols. Additionally, PLOS ONE offers an option for publishing peer-reviewed Lab Protocol articles, which describe protocols hosted on protocols.io. Read more information on sharing protocols at https://plos.org/protocols?utm_medium=editorial-email&utm_source=authorletters&utm_campaign=protocols.

We look forward to receiving your revised manuscript.

Kind regards,

Pracheth Raghuveer, MD, DNB

Academic Editor

PLOS ONE

Journal Requirements:

Additional Editor Comments (if provided):

Reviewers' comments:

Reviewer's Responses to Questions

**Comments to the Author**

1. If the authors have adequately addressed your comments raised in a previous round of review and you feel that this manuscript is now acceptable for publication, you may indicate that here to bypass the “Comments to the Author” section, enter your conflict of interest statement in the “Confidential to Editor” section, and submit your "Accept" recommendation.

Reviewer #2: All comments have been addressed

Reviewer #3: All comments have been addressed

2. Is the manuscript technically sound, and do the data support the conclusions?

Reviewer #2: Yes

Reviewer #3: Yes

3. Has the statistical analysis been performed appropriately and rigorously? 

Reviewer #2: N/A

Reviewer #3: Yes

4. Have the authors made all data underlying the findings in their manuscript fully available?

Reviewer #2: (No Response)

Reviewer #3: Yes

5. Is the manuscript presented in an intelligible fashion and written in standard English?

Reviewer #2: Yes

Reviewer #3: Yes

6. Review Comments to the Author

Reviewer #2: The authors have addressed all the reviewers comments. Now the manuscript is accepted in its present form.

Reviewer #3: All figures must be reinserted in a high quality format. They are all blur. Authors achieved a good

7. PLOS authors have the option to publish the peer review history of their article (what does this mean?). If published, this will include your full peer review and any attached files.

Reviewer #2: No

Reviewer #3: No

---

## [Author Response · Author response to Decision Letter 1]

23 Feb 2024

There were no specific reviewer and editor comments in the decision letter. However, there were general instructions regarding formatting of figures (via PACE diagnostic tool), checking of references for retractions and so on. These instructions have been followed to the best of our ability.

---

## [Editor Report · Decision Letter 2]

19 Mar 2024

Readiness of primary care centres for a community-based intervention to prevent and control noncommunicable diseases in the Caribbean: a participatory, mixed-methods study

PONE-D-22-23088R2

Dear Dr. Goberdhan,

We’re pleased to inform you that your manuscript has been judged scientifically suitable for publication and will be formally accepted for publication once it meets all outstanding technical requirements.

Kind regards,

Pracheth Raghuveer, MD, DNB

Academic Editor

PLOS ONE